# GUIDE: Real-Time Human-Shaped Agents

**Lingyu Zhang[1], Zhengran Ji[1], Nicholas R Waytowich[2], Boyuan Chen[1]**
[1]Duke University, [2]Army Research Laboratory
http://www.generalroboticslab.com/GUIDE

## Abstract

The recent rapid advancement of machine learning has been driven by increasingly powerful models with the growing availability of training data and computational resources. However, real-time decision-making tasks with limited time and sparse learning signals remain challenging. One way of improving the learning speed and performance of these agents is to leverage human guidance. In this work, we introduce GUIDE, a framework for real-time human-guided reinforcement learning by enabling continuous human feedback and grounding such feedback into dense rewards to accelerate policy learning. Additionally, our method features a simulated feedback module that learns and replicates human feedback patterns in an online fashion, effectively reducing the need for human input while allowing continual training. We demonstrate the performance of our framework on challenging tasks with sparse rewards and visual observations. Our human study involving 50 subjects offers strong quantitative and qualitative evidence of the effectiveness of our approach. With only 10 minutes of human feedback, our algorithm achieves up to 30% increase in success rate compared to its RL baseline.

## 1 Introduction

Many real-world tasks are real-time decision-making processes [7, 48] that require understanding the problem, exploring different options, making decisions, refining our understanding, and adjusting decisions accordingly. Due to their inherent complexity, these tasks pose significant challenges for current machine learning systems. Effective exploration[8, 16, 41, 36] is crucial for gathering informative data, especially in mission-critical tasks such as search and rescue, disaster response, and medical emergencies where time is limited. Furthermore, open-world problems often lack dense labels and provide extremely sparse environment feedback signals[42, 19, 29, 21]. Therefore, agents must reason over long horizons to make informed decisions [24, 26, 43].

Human-guided machine learning[32, 33, 34, 31, 1, 52], also known as human-in-the-loop machine learning, has been proposed to integrate human feedback into reinforcement learning (RL) agents. Various methods differ in how they obtain, format, and incorporate human feedback. Some approaches rely on full demonstrations[32, 34] or partial corrections[12] through imitation learning [32, 33, 34] or inverse RL [31, 1, 52], assuming that humans can directly control the agents and possess expert-level task knowledge. Conversely, non-expert humans can often still judge the quality of the agent strategies, leading to comparison-based feedback, such as preferences[2, 20, 45, 18, 46, 17, 47, 23, 53] and rankings [9, 10]. These offline methods, however, require large datasets and parallel prompts and, hence, cannot support real-time guidance, limiting their applicability in dynamic environments.

Discrete label feedback has shown promise for real-time human-guided RL, where humans provide scalar rewards[28, 4] or discrete feedback[25, 44] (e.g., good, neutral, bad). Despite its promise, this approach is constrained by the need for significant human efforts and the less informative nature of

38th Conference on Neural Information Processing Systems (NeurIPS 2024).

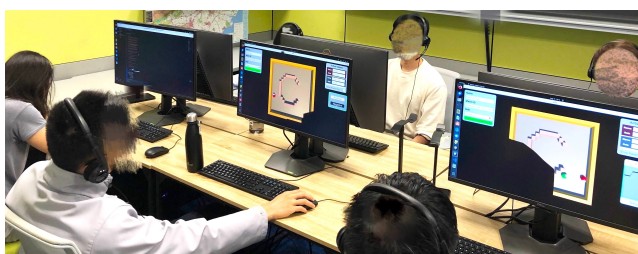

Fig. 1: We propose **GUIDE** as a novel framework for real-time human-shaped agents enabling continuous feedback and continual improvements without human trainers. We also aim to understand how individual differences affect their guided agents' performances.

the discrete feedback. Current research has demonstrated success[25, 44, 28, 4] primarily in simple, low-dimensional tasks with limited solution spaces. Moreover, high-quality human feedback is challenging [11] to obtain due to variability in human guidance skills, cognitive biases, and individual differences. Most studies [25, 44, 28, 4, 50] involve small sample sizes ($N < 10$), often including the designers themselves, which makes it difficult to quantitatively assess the applicability of the methods to the real world. Furthermore, how to keep refining the agent without continuous human input remains unclear.

We introduce GUIDE (Fig. 1), a framework for real-time human-guided RL that enables continuous human feedback and grounds such feedback into dense rewards, thereby accelerating policy learning. GUIDE also includes a parallel training algorithm that learns a simulated human feedback model to effectively reduce human inputs and enable continued policy improvements without human feedback. Our experiments span three challenging tasks characterized by continuous action spaces, high-dimensional visual observations, temporal and spatial reasoning, multi-agent interactions, and sparse environment rewards. Our human studies involving 50 participants provide strong quantitative and qualitative support to the efficacy of our approach. Additionally, we conduct a series of cognitive tests and analyses to quantify individual differences among participants and explore how these differences correlate with agent learning performance. Our further analysis of the learned policies highlights the critical role of alignment in developing effective human-guided agents.

## 2 Related Work

**Human-Guided Machine Learning** Various computational frameworks have been proposed to integrate human feedback in guiding machine learning agents. Behavior cloning [32, 33, 34] trains a policy through supervised learning on human demonstrations assuming direct human control of the agents. Inverse reinforcement learning [31, 1, 52] infers objectives from human demonstrations for police optimization, but it requires extensive human demonstrations and often struggles with accurate objective inference and scalability. Human preference-based learning [2, 20, 45, 18, 46, 17, 47, 23] allows humans to score trajectories through comparisons, reducing the need for full demonstrations yet still depending on offline policy sampling. GUIDE differs by focusing on real-time decision-making tasks without relying on abundant offline datasets or parallel trajectory rollouts.

**Real-Time Human-Guided Reinforcement Learning** Our work is formalized into an RL framework to integrate real-time human feedback into the policy learning process. TAMER [25] learns to regress a reward model from human feedback, which is used as a typical reward function for policy learning. Deep TAMER [44] generalizes this approach to higher dimensional inputs and uses a neural network to parameterize both the policy and the reward model. COACH [28] and Deep COACH [4] address the inconsistency of human rewards observed during training by modeling human feedback as an advantage function in an actor-critic framework. Deep COACH scales COACH to handle higher-dimensional observations and more expressive policy representations. Recent advancements have further improved these algorithms, addressing various challenges in ground human feedback, such as different feedback modalities, stochasticity of feedback [3], and continuous action space [37].

However, existing real-time human-guided RL studies share several limitations. Simplifying human feedback to discrete sets {positive, neutral, negative} with rewards $\{1, 0, -1\}$ requires users to focus on clicking or typing the correct options which can be distracting. Such simplification also causes information loss by ignoring intermediate values. This approach also introduces hyperparameters to

associate feedback to trajectories in time, requiring extensive tuning. Moreover, most studies validate their methods on simple tasks with discrete actions and low-dimensional observations. Furthermore, human studies are often limited to the designers themselves, a small group of subjects ($N < 10$, mostly $N < 5$), or a trained expert policy with RL to simulate humans. It remains unclear whether the existing approaches scale to general populations. There is also a large gap in understanding how individual differences among human participants affect policy training. Lastly, there is little discussion on continuing to improve policies in the absence of human feedback.

**Key Novelties of GUIDE** Unlike previous approaches, GUIDE enables continuous human feedback, maximizing guidance information with minimal hyperparameter tuning. Our experiments cover challenging tasks with continuous action spaces, high-dimensional visual observations, spatial-temporal reasoning, and multi-agent interactions. GUIDE is also evaluated on the largest sample of human subjects ($N = 50$) among related studies, incorporating cognitive assessments to characterize individual differences and their impact on guiding learning agents. Additionally, GUIDE includes a parallel training algorithm to mimic human feedback for continued policy improvements when direct human feedback becomes unavailable.

## 3 Preliminary

### 3.1 Value-Based Reinforcement Learning

Value-based Reinforcement Learning is an effective approach to solving decision-making problems. In value-based RL, a key element is the action-value function $Q(s, a)$ which provides an estimate of the total expected return when taking an action $a$ in state $s$ and following a specific policy thereafter [40]. The policy $\pi$ is derived from $Q$ by selecting the action that maximizes the action-value function:

$$\pi(s) = \underset{a}{\operatorname{argmax}} \, Q(s, a) \qquad (1)$$

The action-value function $Q(s, a)$ at time step $t$ can be defined as the expected sum of discounted future rewards with discounted factor $\gamma$:

$$Q(s, a) = \mathbb{E}[\sum_{k=0}^{\infty} \gamma^k R_{t+k+1} | s, a] \qquad (2)$$

Where $R_{t+k+1}$ is the reward $k + 1$ steps after the current step. In valued-based deep reinforcement learning, a neural network is used to approximate the action-value function to tackle challenges such as high-dimensional and partial observations and complex decision boundaries. The training of these networks typically involves a combination of Monte-Carlo sampling and bootstrapping techniques, which leverage current estimates of $Q$ to improve future predictions. The accuracy of the $Q$-function is crucial, especially in environments where reward signals are sparse and delayed, posing significant challenges to the efficiency and efficacy of the learning process.

### 3.2 Human feedback as Value Functions

A natural way for humans to guide an agent's learning is to observe its inputs and provide feedback on its actions. This translates directly to incorporating human feedback into reinforcement learning by assigning human feedback as the myopic state-action value. Deep TAMER [44] is a prominent human-guided RL framework that leverages this concept by enabling humans to offer discrete, time-stepped positive or negative feedback. To account for human reaction time, a credit assignment mechanism maps feedback to a window of state-action pairs. A neural network $F$ is trained to estimate the human feedback value, denoted as $\hat{f}_{s,a}$, for a given state-action pair $(s, a)$. The policy then selects the action that maximizes this predicted feedback.

We constructed a strong baseline by enhancing Deep TAMER in numerous ways. First, the original Deep TAMER's greedy action selection method only works with discrete action spaces. We designed a continuous version of Deep TAMER while adopting state-of-the-art reinforcement learning implementation practices. We implement an actor-critic framework to handle continuous action space. Here, the critic is the human feedback estimator parameterized by $\phi$: $F_\phi(s, a)$, directly estimating human feedback instead of a target action value. The actor is parameterized by $\theta$: $A_\theta(s) = a$, aiming

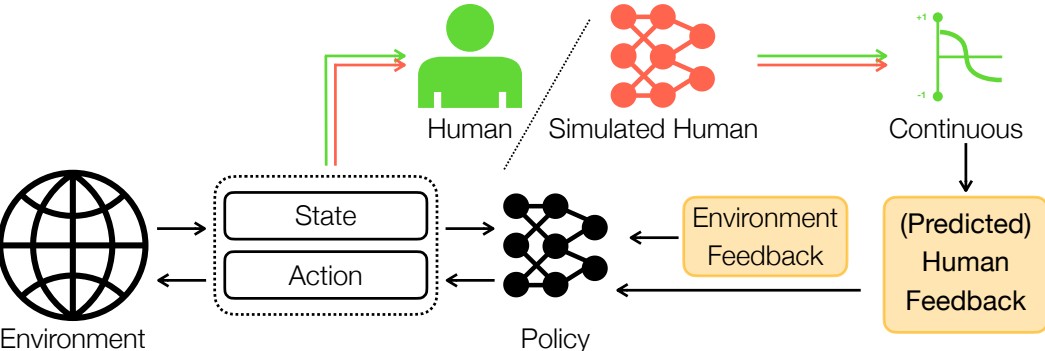

Fig. 2: **GUIDE:** The training consists of two stages: During the Human guidance stage, the human trainer observes the state and action taken by the agent and provides real-time continuous feedback. The feedback values are grounded into per-step dense rewards and combined with the environment reward. Concurrently, we train a human feedback simulator that takes in state-action pairs and regresses the feedback values. During the Automated guidance stage, the trained simulator stands in for the human and provides feedback to continue to improve the policy, effectively reducing human efforts and cognitive loads.

to maximize the critic's output. The combined objective is defined as $\mathcal{L}_{\text{c-Deep TAMER}} = \mathcal{L}_\theta + \mathcal{L}_\phi$, where:

$$\mathcal{L}_\theta = -F_\phi(s, A_\theta(s)) \tag{3}$$

$$\mathcal{L}_\phi = ||F_\phi(s, A(s)) - f_{s, A_\theta(s)}||_2 \tag{4}$$

We follow recent advancements in model architectures and optimization strategies, such as target net soft updates and using Adam optimizer instead of SGD. Our strong baseline not only enhances Deep TAMER to continuous actions and recent RL practices but also maintains the core methodology of integrating real-time human feedback into the learning process.

## 4 GUIDE: Grounding Real-Time Human Feedback

### 4.1 Method Overview

Developing real-time decision-making agents presents significant challenges. These agents must explore high-dimensional observations, constantly adapt, and make accurate decisions with limited supervisory signals. Real-time human-guided RL seeks to harness human expertise and adaptability. However, existing algorithms often underutilize human expertise by solely relying on discrete feedback. Additionally, integrating human feedback with environmental rewards can be complex, and human evaluation criteria often evolve during training.

This section introduces our framework **GUIDE** (Fig. 2) to address these limitations. GUIDE enables real-time human guidance using dense and continuous feedback. The agent directly perceives the environment through visual frames, selects and executes actions, and receives continuous human feedback per decision step. This feedback is then converted into dense rewards that will be integrated with sparse environment rewards for effective learning. Concurrently, a separate neural network consistently observes the state-action pairs and human feedback and learns to predict the provided human feedback. This learned model will eventually replace the human as the feedback provider, significantly improving the efficiency of human-guided RL while minimizing the required human guidance time.

### 4.2 Obtaining Continuous Human Feedback

Conventional real-time human-guided RL frameworks rely on feedback that is discrete in time and values. An example that we adopted as our baseline is shown in Fig. 3(A). We propose a novel interface that enables human trainers to provide continuous-valued feedback at every time step. As illustrated in Fig. 3(B), the trainer hovers their mouse over a window to indicate their assessment of the agent's behavior.

Our method offers several advantages:

- **Natural Engagement:** Hovering is a more natural interaction compared to clicking buttons [39], fostering continuous human guidance without interrupting the training flow.
- **Expressive Feedback:** Continuous values allow humans to express nuanced assessments, capturing a wider range of feedback compared to discrete options, as shown in Fig. 3.
- **Constant Training:** Unlike discrete-feedback algorithms where the model update frequency is based on the feedback frequency, the continuous feedback ensures that the model continuously learns and adapts.
- **Simplified Credit Assignment:** In our setting, it is reasonable to assume a constant human feedback delay, alleviating the need for complex credit assignment to map delayed feedback to specific state-action pairs. Such delayed association time window has been reported to have inconsistent values across different studies. In contrast, we used the same one-time delay factor for all our studies.

### 4.3 Learning from Joint Human and Sparse Environment Feedback

While human guidance offers valuable supervision, sparse environment rewards or terminal rewards remain crucial for their easy access and the ability to induce desired targets.

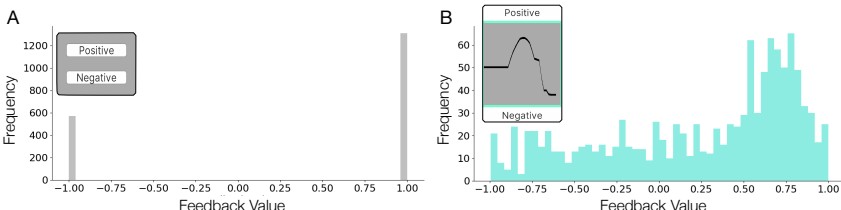

Fig. 3: (A) Conventional discrete feedback. (B) Our continuous feedback. The histograms indicate the feedback distribution provided by the same subject on the same task. Continuous feedback carries more information from the human trainer.

However, directly using human feedback as a value function makes incorporating existing environment rewards challenging [44]. We propose a straightforward but effective solution: convert the human feedback at each time step $t$ into a reward, denoted as $f_t = r_t^{hf}$. This allows the seamless integration of environment rewards through simple addition: $r_t = r_t^{hf} + r_t^{env}$. This approach also enables leveraging off-the-shelf advanced RL algorithms, which typically make minimal assumptions about the reward function. Our method can be seen as interactive reward shaping [30] which is a vastly effective approach in handling sparse and long-horizon tasks. However, designing dense reward functions directly can be difficult as it often requires significant prior knowledge, manual effort, and trial and error. Moreover, it lacks adaptability to unforeseen scenarios. Our approach allows for real-time reward shaping through human guidance, capitalizing on human flexibility and intuition.

### 4.4 Continue Training by Learning to Mimic Human Feedback

Human guidance can be time-consuming and cognitively demanding. To maximize the benefit of human input while minimizing their effort, we propose a regression model that learns to mimic human feedback, acting as a surrogate for the human trainer (Fig. 2). This model allows for continual policy improvements even when human feedback is not available. Our idea is based on the assumption that human feedback implicitly follows a mapping function: $H(s, a) = f$, where $s$ and $a$ are the state and action observed by the human, and $f$ is the assigned feedback value.

The key insight is that during human guidance, we can readily collect state-action pairs along with their assigned feedback values, which allows training a human feedback simulator, parameterized by a neural network $\hat{H}(s, a) = \hat{f}$, by minimizing $||f - \hat{f}||_2$. This training can occur concurrently with human guidance to prepare to substitute at any time or offline for later deployment in continual training. To prevent overfitting, we held out 1 out of 5 trajectories as a validation set. Once trained, the simulator can operate in inference mode to provide feedback in the absence of the human trainer. This model makes no assumptions about the human's specific feedback style or patterns. Instead, it learns to provide feedback consistent with the human trainer, minimizing the shift in reward distribution.

# 5 Experiments

## 5.1 Environments

We conduct our experiments on the CREW [51] platform.

**Bowling:** A modified version of Atari Bowling. Each episode consists of 10 rolls. The agent receives a reward equal to the number of pins hit after each roll. The action space is 3-dimensional: the initial ball position, the distance to start steering the ball, and the steer direction. Observation is the visual image.

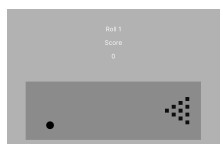

**Find Treasure:** The agent is tasked to navigate through a partially observable maze to retrieve a treasure. The agents receive a +10 reward upon reaching the treasure, and a constant -1 time penalty for each step taken. The observation space is a top-down accumulated view of where the agent has explored, initialized by a square area around the agent. The action space is a two-dimensional vector of the next destiny location, to which a low-level planning algorithm will navigate the agent. The max episode length is 15 seconds.

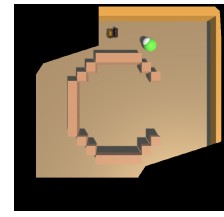

**1v1 Hide-and-Seek:** A multi-agent one-on-one visual hide-and-seek task [13, 14], where we aim to learn the seeker policy whose goal is to navigate through the maze and catch the hider. The hider policy is a heuristic that will avoid obstacles and reflect when running into a wall while moving away from the seeker if the seeker is within a certain range. The reward, observation, and action spaces remain the same as Find Treasure.

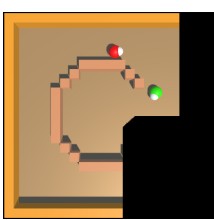

## 5.2 Experiment Settings

**Evaluation Metrics** We allocated 10 minutes for consistent human feedback. We collected the checkpoints and human feedback data and performed continual training for another 10 minutes. The model's performance was evaluated in a test environment every minute for Bowling and every 2 minutes for Find Treasure and Hide-and-Seek. For every Bowling checkpoint, we evaluate for 1 game (10 rolls); for Find Treasure and 1v1 Hide-and-Seek, we evaluate 100 episodes for every checkpoint. All test results are reported on unseen test conditions.

**Baselines** We include two state-of-the-art reinforcement learning algorithms and a human-guided RL algorithm as baselines. We selected Deep Deterministic Policy Gradient (DDPG) [27] and Soft Actor-Critic (SAC) [22] due to their superior performance on many benchmarks. We compared GUIDE to the continuous version of Deep TAMER as described in Sec. 3.2, denoted as c-Deep TAMER under the same amount of human training time. We also run experiments on Find Treasure and Hide and Seek with heuristic feedback, in other words, a dense reward. The feedback is an exploration reward when the target (treasure or hider) is not within the agent's visible view and a distance reward when it is. This can be seen as an upper bound of our method, as this is the "heavy engineered reward" by domain experts with no noise or delay.

**Human Subjects.** We recruited 50 adult subjects to participate in the experiments under the approval of the Institutional Review Board. The subjects have no prior training or knowledge of the algorithms or tasks and have only been instructed to assess the performance of the agents. Inter-trial intervals are included between individual game experiments. For each session, the subject will receive $20 compensation.

## 5.3 Human Cognitive Tests for Individual Difference Characterization

In order to quantify individual differences among participants and explore how these differences affect the human-guided RL performance, we conducted a series of cognitive tests to evaluate their cognitive abilities relevant to the training tasks before the human-guided RL experiments. We describe the details of the cognitive tests as follows. All tests involve six trials for each participant.

**Eye Alignment [35](Fig. 4)(A):** The subject is asked to align a ball on the left side of the screen with a target on the right side as accurately as possible within five seconds without using any tools.

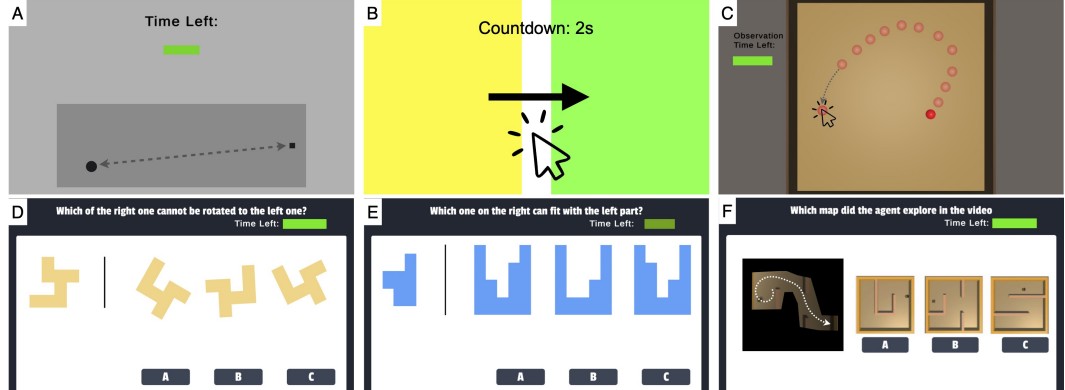

Fig. 4: **Cognitive Tests:** We conducted a series of cognitive tests to quantify how individual differences among subjects affect their guided agents' performances. (A) Eye Alignment (B) Reflex (C) Theory of Behavior (D) Mental Rotation (E) Mental Fitting (F) Spatial Mapping

The score is calculated as the negative average of the distances between the ball's and the target's horizontal positions across trials.

**Reflex [6](Fig. 4)(B):** The subject is asked to click the screen as quickly as possible when it changes from yellow to green. Clicking while the screen is still yellow or failing to click within two seconds after the screen turns green is considered a failure. The score is calculated as the negative average of the reflex times, with failures being left out in the score calculation.

**Theory of Behavior [15](Fig. 4)(C):** The subject observes a red ball moving in an unknown but fixed pattern for five seconds. Afterward, the ball pauses, and the subject must predict the ball's position one second after it resumes moving. The subject has two seconds to make this prediction. The score is calculated as the negative average of the distances between the ball's actual positions and the subject's predicted positions across trials.

**Mental Rotation[38](Fig. 4)(D):** The subject is tasked with identifying the piece among three similar pieces that cannot be directly rotated to match the target piece within eight seconds. The score is calculated based on the accuracy of the subject's identifications across all trails.

**Mental Fitting[38](Fig. 4)(E):** The subject is tasked with identifying the only piece among three similar pieces that can fit with the target piece within eight seconds. The score is calculated based on the accuracy of the subject's identifications across all trials.

**Spatial Mapping [5](Fig. 4)(F):** A video of an agent navigating a maze with a restricted field of view is presented to the subject. After watching the video, the subject is asked to identify the maze from a selection of three similar mazes within five seconds. The score is calculated based on the accuracy of the subject's identifications across all trials.

### 5.4 Implementation Details

**RL backbone and model architecture.** The state-of-the-art visual reinforcement learning algorithm transitioned from SAC [22] to DDPG [27] due to its stronger performance enabled by the easy incorporation of multi-step returns and the avoidance of entropy collapse[49]. Following this, we also select DDPG [27] as our Reinforcement Learning backbone for GUIDE. We used a 3-layer convolutional neural network as the vision encoder. The actor, critic, and learned feedback model shared this encoder, and each follows it with a 3-layer Multi-Layer Perceptron.

**Hyperparameters** We used an Adam optimizer with a fixed learning rate of 1e-4 for RL policy training, with a discount factor of $\gamma = 0.99$. We applied gradient clipping setting max grad norm to 1. For the learned feedback model, we used the same Adam optimizer with 1e-4 learning rate and employed early stopping based on the loss on held-out trajectories. For Deep TAMER's credit assignment window, we used the same uniform $[0.2, 4]$ distribution as in the original paper. We used

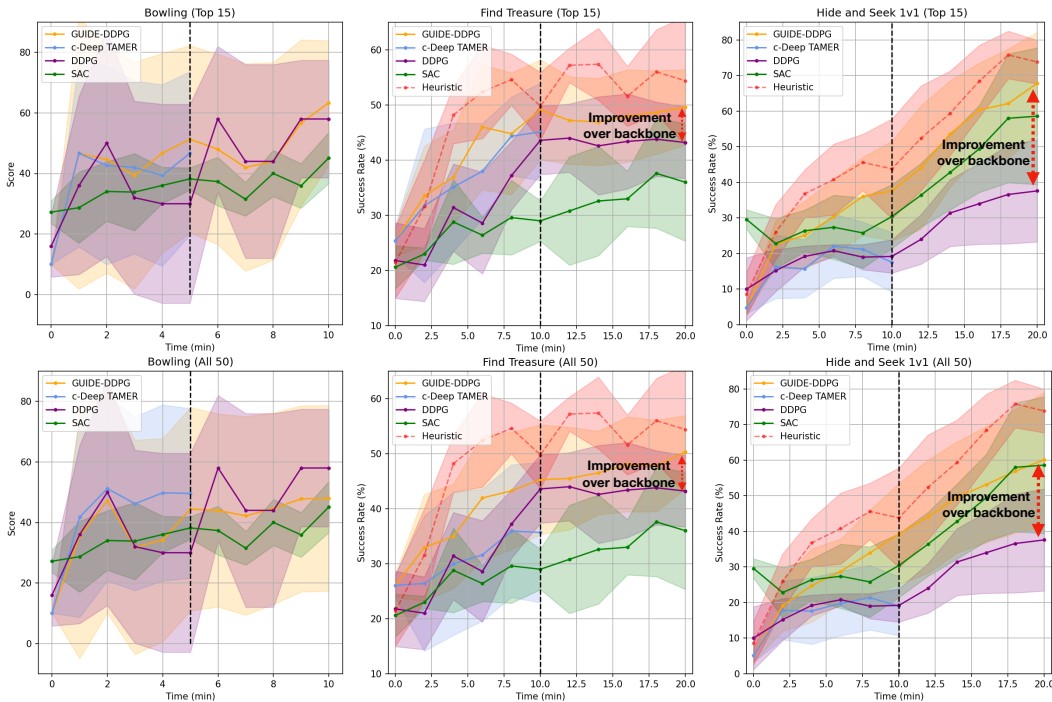

Fig. 5: GUIDE performance compared with other baselines. In challenging tasks, GUIDE consistently outperforms all other baselines. Subjects with higher cognitive test scores also result in higher performance in the learned agents as shown in the top row (Top 15).

a shorter window of $[0.2, 1]$ for Find treasure and Hide-and-Seek. For these more difficult navigation tasks, we stacked three consecutive frames as input.

**Human delay factor.** Human delay is non-neglectable in human-guided RL. The time it takes for a human to see and process the states and to produce a feedback signal varies across environments and individuals. Preliminary experiments with the authors suggested that shifting human feedback by 2 seconds for Bowling and 1 second for Find Treasure and Hide and Seek best aligned with the intended state-action pairs.

## 5.5 Experiment Results and Analysis

**Human-Guided RL performance** We report our results in Fig. 5, where the x-axis is the total training time, and the y-axis is either the score for Bowling or the success rate for Find Treasure and Hide-and-Seek. The dashed line separating the first and second 10 minutes indicates the end of the human guidance and the beginning of the automated guidance. The first row reports the training results of the 15 human trainers who scored the highest on the cognitive test. The second row is the average performance of all 50 human subjects. We observe that subjects who tested higher on cognitive skills result in higher performing agents, as the trained AI performance approaches closer to the heuristic feedback upper bound. For the simple Bowling task, all methods perform similarly, with GUIDE having a slight advantage over DDPG. On the challenging Find Treasure and Hide-and-Seek task, GUIDE-DDPG scored up to 30 percent higher success rate than its RL counterpart, and 50 percent higher than c-Deep TAMER given the same amount of human guidance time. It is worth noting that in this time-critical setting, GUIDE-DDPG is able to reach the same level of performance as its RL baselines within half the training time. The average of all 50 untrained human trainers was able to surpass its RL baseline by a large margin while also surpassing our greatly enhanced Deep TAMER on the challenging Find Treasure and Hide and Seek by up to 40 points in success rate.

**Exploration Analysis** The ability to explore the state space is considered one of the key aspects of successful sparse reward RL. We hypothesize one benefit of human guidance is inducing efficient exploration behavior. We quantify exploration tendency on Find Treasure by measuring the accumulated visible area ratio. After two minutes of human feedback within the GUIDE framework, the

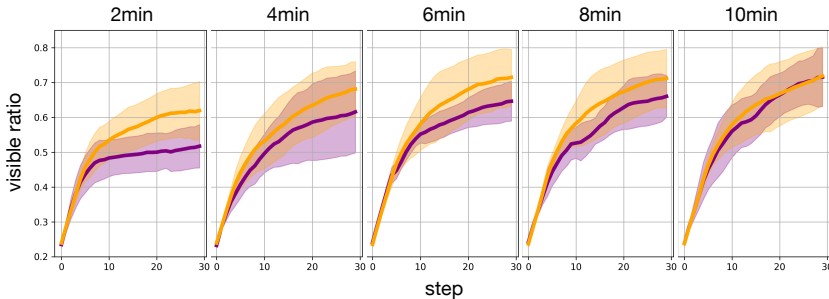

Fig. 6: Exploration behavior of GUIDE and DDPG agents. For each of the plots, the x-axis is the step number through the course of an episode. The y-axis is the ratio between the area of the visible view and the entire input frame. We observe a stronger tendency of exploration exhibited by the human-guided agent compared to the baseline RL agent.

agents have shown more efficient exploration behavior compared to the RL baseline. As shown in Fig. 6, each plot corresponds to one stage in training. For each plot, the x-axis is the step number through the course of an episode, and the y-axis is the visible area ratio for that step averaged across 10 episodes. The GUIDE results are tested on the human subjects who ranked top 15 in cognitive tests. The DDPG results are tested on five different random seeds. The higher the curve, the faster the agent explores the maze. We find that the exploration ability increases throughout training for both DDPG and GUIDE, with GUIDE learning to explore faster in the first 10 minutes. As the training progresses, exploration reaches its bottleneck utility, and DDPG and GUIDE converge to a similar exploration rate.

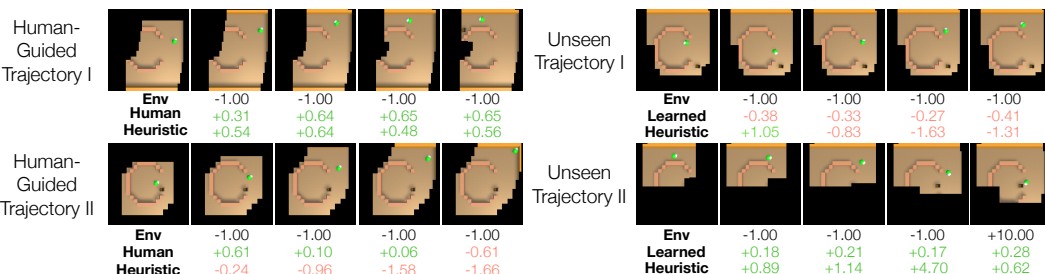

Fig. 7: Qualitative visualization of the learned human feedback model. Our learned feedback model is able to generalize to unseen trajectories and provide effective feedback in place of humans.

**Learned Feedback Model Ablations** We qualitatively analyze the effect of the learned feedback model by visualizing its predictions on unseen trajectories. As shown in Fig. 7, our learned feedback model generalizes to unseen trajectories and is able to provide effective feedback in place of humans.

**Individual Differences vs. Performance** To further characterize individual differences and how they affect policy training, we found a significant correlation between the cognitive test ranking of the human subjects and their trained AI performance, with a Pearson correlation coefficient of 7.522 and a p-value of 0.001, as shown in Fig. 8. Among all cognitive tests, the mental rotation test and Mental Fitting test are the strongest indicators of overall guided AI success. We also discovered relationships between individual tasks and cognitive tests. Notably, 1v1 Hide-and-Seek score is positively related to reflex time. This is likely due to the fact that Hide-and-Seek is a quickly evolving dynamic task, and faster response time will

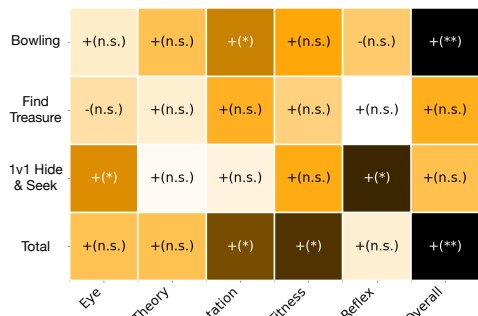

Fig. 8: Correlation between cognitive test scores (normalized) and GUIDE training performance. The darker the color, the more statistically significant the correlation. "+": positive, "-": negative.

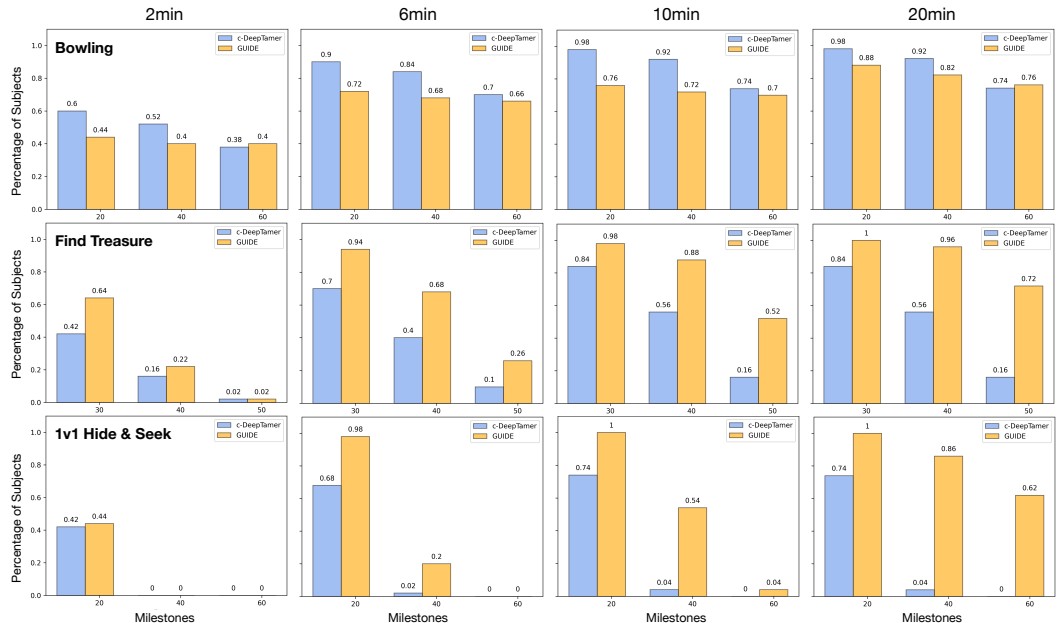

Fig. 9: Robustness to individual differences. The subplots from left to right show how performance evolves over training time. In each subplot, the x-axis is training milestones indicated by task scores, and the y-axis is the percentage of humans who were able to train the agent to reach the milestone at a given time. We find that GUIDE's robustness scales to more complex tasks.

result in better matching in feedback and state-action pairs. Detailed correlation results can be found in the appendices.

**Robustness to individual differences** A critical aspect of human-guided machine algorithms is the method's robustness to different human trainers. A generalizable algorithm should maintain a strong performance when human feedback patterns vary. However, this is rarely discussed by prior methods likely due to the small size of human participants. Here we provide a thorough analysis on GUIDE's performance across our 50 human trainers compared to c-Deep TAMER. Each subplot in Fig 9 shows for each human-guided algorithm, the percentage of human trainers (y-axis) that are able to train the agent to reach specific milestones indicated by task scores (x-axis). Panel A shows that for the Bowling task, c-Deep TAMER and GUIDE are close in performance. Panel B and C shows that for Find Treasure and 1v1 Hide-and-Seek, as the complexity of the game increases, more human trainers are able to reach milestones using GUIDE than using c-Deep TAMER, demonstrating the scalability of the robustness of GUIDE to individual differences.

## 6  Conclusion, Limitations and Future Work

We introduce GUIDE, a novel framework for real-time human-guided RL. GUIDE leverages continuous human feedback to effectively address limitations associated with sparse environment rewards and discrete feedback approaches. Furthermore, GUIDE incorporates a parallel training algorithm to learn a human feedback simulator, enabling continued policy improvements without ongoing human input. Our extensive experiments demonstrate the efficacy of GUIDE across challenging tasks while presenting human cognitive assessments to understand individual differences.

While GUIDE offers significant advantages, it presents some limitations and exciting avenues for future work. First, our current evaluation focuses on tasks with moderate complexity. Future work can explore scaling GUIDE to highly complex environments and large-scale deployments. Moreover, we have quantitatively observed strong individual differences in guiding learning agents through extensive human studies. However, we have not explored how to mitigate such differences and account for human variability in feedback patterns. Furthermore, understanding how the human feedback simulator operates and interprets the agent's behavior remains an open question. Future work will delve into explainable learning techniques to shed light on this process.

# 7 Acknowledgements

This work is supported in part by ARL STRONG program under awards W911NF2320182 and W911NF2220113.

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

# A Computatonal Resources

All human subject experiments are conducted on desktops with one NVIDIA RTX 4080 GPU. All evaluations are run on a headless server with $8 \times$ NVIDIA RTX A6000 and NVIDIA RTX 3090 Ti.

# B Fully Cognitive test-Guided AI performance Study

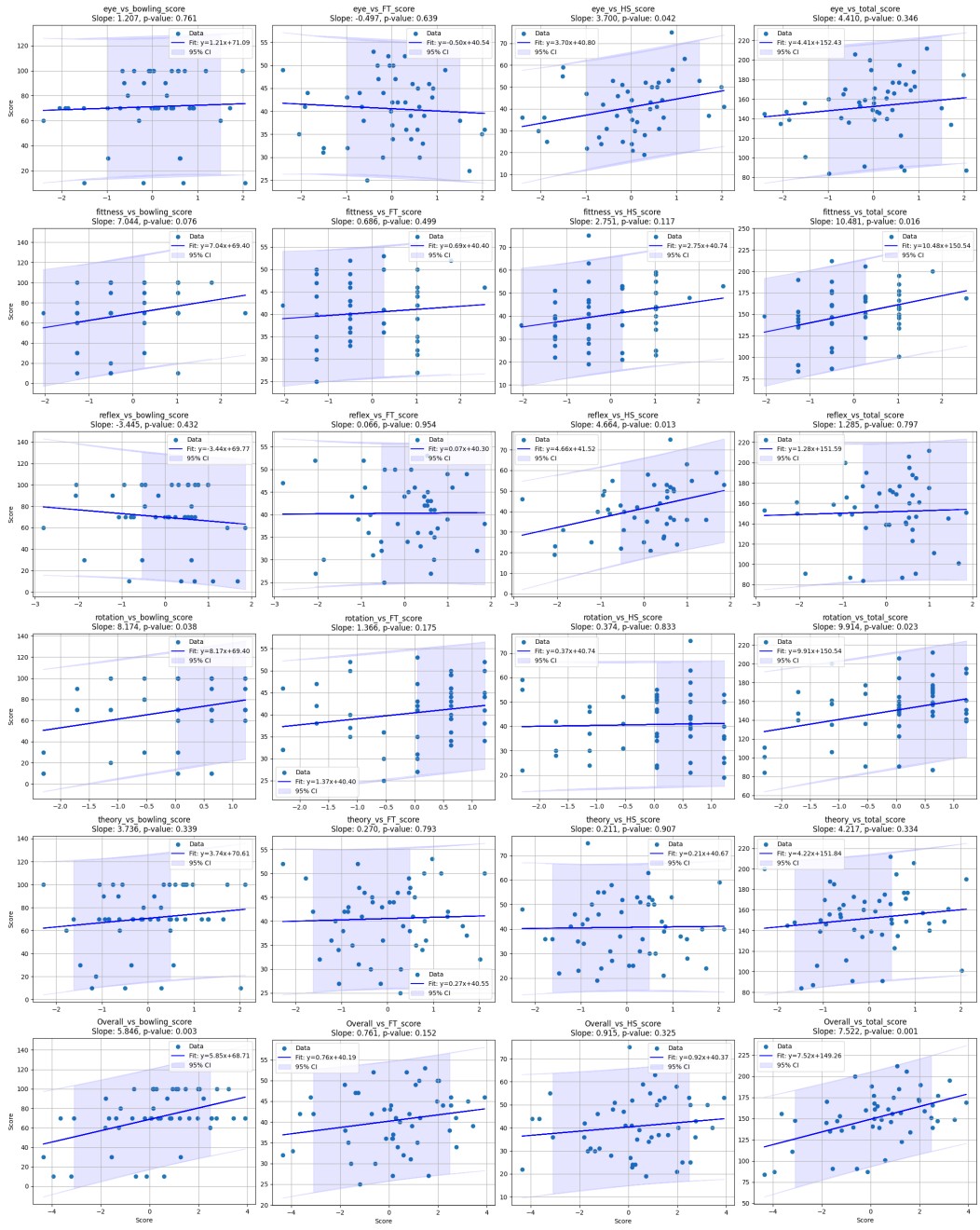

