# OpenReview forum: "GUIDE: Real-Time Human-Shaped Agents"
_NeurIPS.cc/2024/Conference — NeurIPS 2024 poster_

### Official Review · Reviewer_JtyH · 2024-06-27

**Soundness:** 3
**Presentation:** 2
**Contribution:** 2
**Rating:** 3
**Confidence:** 4

**Summary:**

The paper introduce GUIDE, a RLHF framework for real-time RLHF with online and continous human feedback. GUIDE translates the human feedback to dense reward. Addtionally, GUIDE includes a parallel training model that learns a simulated human feedback. By involving 50 participants annotation, GUIDE solves three typical challenging sparse reward environments.

**Strengths:**

1. The paper is easy to read.
2. GUIDE firstly proposed novel continous human feedback and is also evaluated human annotators.
3. GUIDE demonstrated improvement compared to baselines in 3 environments, and analyzed cognitive tests and analyses.

**Weaknesses:**

1. My biggest concern comes from the practicality of GUIDE. From both theoretical and experimental perspectives, I find it hard to believe that such a simple continuous feedback model can be applied to real-world scenarios. For example, the paper states in line 38 that "Current research has demonstrated success primarily in simple, low-dimensional tasks with limited solution spaces." However, the experiments conducted in the paper also involve environments where baseline algorithms like DDPG or SAC can converge with a good reward function after only about **10 minutes of training**. Moreover, according to the experimental results, GUIDE, which incurs a high cost of human feedback, does not outperform manually designed simple rewards (such as the distance to the target, I think it is not hard to design it). Therefore, despite the fact that the environments used do have continuous actions and image inputs, I believe these environments are not suitable for validating RLHF algorithms because the reward functions are easy to design and the tasks themselves are simple.
2. The core argument of the paper is that continuous real-time feedback is extremely difficult to implement in practice. It requires annotators to constantly provide scalar rewards without pause, and such absolute value annotations are more susceptible to biases from different individuals. Pair-wise annotation is much easier than absolute value annotation and can be conducted asynchronously with the training process. If an AI agent needs to be trained for several days, the cost will be unacceptable.
3. Although the paper suggests using model predictions to synthesize feedback, such a simple supervised learning regression objective is unlikely to accurately model the complex reward distribution. My reasoning is that predicting the relative goodness of A and B is easier than predicting scalar reward values, but there will still be many prediction biases.
4. The definitions of various symbols in the paper are imprecise and confusing, for example:
- What is the meaning of A(s, a) in Equation 1? Also, A(s) = a in the line 123, is it the same?
- difference of Q(s, a) and q?
- How to get the r^hf?
These typos make it very difficult to understand the details of the paper.
5. There is a lack of discussion on recent related works in RLHF, such as:
- [1] White D, Wu M, Novoseller E, et al. Rating-based reinforcement learning[C]//Proceedings of the AAAI Conference on Artificial Intelligence. 2024, 38(9): 10207-10215.
- [2] Yuan Y, Hao J, Ma Y, et al. Uni-RLHF: Universal Platform and Benchmark Suite for Reinforcement Learning with Diverse Human Feedback[J]. ICLR2024.
- [3] Guan L, Verma M, Guo S S, et al. Widening the pipeline in human-guided reinforcement learning with explanation and context-aware data augmentation[J]. Advances in Neural Information Processing Systems, 2021, 34: 21885-21897.
- [4] Guan L, Valmeekam K, Kambhampati S. Relative behavioral attributes: Filling the gap between symbolic goal specification and reward learning from human preferences[J]. ICLR2023.

**Questions:**

Can you provide more details about the annotator's annotations, such as the actual interface and the frequency of operations?

**Limitations:**

Yes, and yes.

---

> ### Author Rebuttal · Authors · 2024-08-03
>
> We thank the reviewer for their thoughtful comments. We would like to address all of your concerns and questions below with point responses:
>
> ----
> >Practicality: “I find it hard to believe that such a simple continuous feedback model can be applied to real-world scenarios. For example, the paper states in line 38 that "Current research has demonstrated success primarily in simple, low-dimensional tasks with limited solution spaces." However, the experiments conducted in the paper also involve environments where baseline algorithms like DDPG or SAC can converge with a good reward function after only about 10 minutes of training.”
>
> We respectively disagree with the reviewer on this observation. Fig.5 shows the opposite observation where baseline algorithms did not converge within 10 minutes and showed no sign of convergence in 20 minutes in our new environments, Find Treasure and Hide-and-Seek. Therefore, these tasks remain significantly challenging for the current RL baselines. GUIDE, on the other hand, was able to improve over baselines by a large margin.
>
> >“GUIDE, which incurs a high cost of human feedback, does not outperform manually designed simple rewards (such as the distance to the target, I think it is not hard to design it). Therefore, despite the fact that the environments used do have continuous actions and image inputs, I believe these environments are not suitable for validating RLHF algorithms because the reward functions are easy to design and the tasks themselves are simple.”
>
> As discussed in the paper, hand-design dense rewards typically do not exist in real-world scenarios and require extensive experience in reward engineering and a practical understanding of RL training. However, we aim to enable human guidance in RL for a broader audience who most likely do not have RL experience or reward engineering. Moreover, dense reward designs, such as precise distance to the target, as the reviewer suggested, assume extra information that is not available to GUIDE, whom only perceives partial visual observations. Therefore, our environments are strong and challenging testbeds for developing human-guided RL algorithms. Furthermore, as prior works have only focused on simple bowling games and low-dimensional observations, we believe that our environment is a large step forward in investigating the potential and scalability of human-guided RL.
>
> >“The core argument of the paper is that continuous real-time feedback is extremely difficult to implement in practice. It requires annotators to constantly provide scalar rewards without pause, and such absolute value annotations are more susceptible to biases from different individuals. Pair-wise annotation is much easier than absolute value annotation and can be conducted asynchronously with the training process. If an AI agent needs to be trained for several days, the cost will be unacceptable.”
>
> Our argument is not that continuous real-time feedback is extremely difficult to implement in practice. Instead, we point out that it provides richer feedback and assignments to every state-action pair than previous discrete feedback without a high cognitive load. Our contribution is to enable grounding such novel feedback modality through GUIDE.
>
> As discussed in our paper, while pair-wise annotation is a popular choice in RLHF, it requires parallel policy rollouts in an asynchronous manner or offline trajectory collections. Our setting is different – we focus on real-time decision-making tasks where no such asynchronous rollouts, offline trajectories, or a simulator that can be run multiple times with policy rollouts and resetting are available. Due to the significant difference in the setting, pair-wise annotation is out of scope for our study.
>
> >“Although the paper suggests using model predictions to synthesize feedback, such a simple supervised learning regression objective is unlikely to accurately model the complex reward distribution. My reasoning is that predicting the relative goodness of A and B is easier than predicting scalar reward values, but there will still be many prediction biases.”
>
> As discussed in our paper, while pair-wise annotation is a popular choice in RLHF, such as recent LLM studies, it requires parallel policy rollouts in an asynchronous manner or offline trajectory collections. Our setting is different – we focus on real-time decision-making tasks where no such asynchronous rollouts, offline trajectories, or a simulator that can be run multiple times with policy rollouts and resetting are available. Due to the significant difference in the setting, pair-wise annotation is out of scope for our study.
>
> We agree that more advancements can be made to improve the simulated feedback model. However, prior to our work, there have not been studies enabling continual training in real-time human-guided RL. Our work is the first step towards this with potential future work to improve the optimization designs to handle more complex reward distributions.
>
> >“What is the meaning of A(s, a) in Equation 1? Also, A(s) = a in the line 123, is it the same? Difference of Q(s, a) and q? How to get the r^hf?”
>
> Thank you for catching the typos. We will correct both $A(s, a)$ and $q(s, a)$  to $ Q(s, a)$. $r^{hf}$ is the human feedback reward provided by the human.
>
> >“There is a lack of discussion on recent related works in RLHF, such as: …”
>
> We thank the reviewer for pointing out relevant literature, and we will include these in our revised version.
>
> >Can you provide more details about the annotator's annotations, such as the actual interface and the frequency of operations?
>
> Yes. A screenshot of our interface is shown in Fig3 (B). The user hovers their mouse over the window to provide feedback. Moving upwards indicates stronger positive feedback, and downwards indicates stronger negative feedback. The decision frequency of the games is set to 0.5s/step. Hence, human feedback values are collected every 0.5 seconds.

---

> ### Author Response · Authors · 2024-08-07
>
> Dear Reviewer,
>
> Thank you again for your detailed review of our paper. We aim to try our best to address all your concerns with our point responses above.
>
> We truly value your feedback. As the end of the rebuttal period is approaching, please feel free to let us know if you have any additional questions or comments. We would be happy to answer them. We look forward to hearing your future thoughts!
>
> Best regards,
>
> Authors

---

> > ### Comment · Reviewer_JtyH · 2024-08-08
> >
> > Thank you for your reply, I still think that the GUIDE method is difficult to apply to reinforcement learning agent that require long term training (e.g., a few days, atari/procegen/minecraft), and even if it could be used, sustained manual loads for a few days would still be excessively costly and inconsistent. I have some concerns about GUIDE's usability for real-world tasks, and a few of the toy experiments in the paper don't support the author's point of view. Thanks a lot, I will clearly keep my score.

---

> > > ### Author Response · Authors · 2024-08-08
> > >
> > > Dear reviewer, thank you for your response. We believe there is clearly a misunderstanding of our work.
> > >
> > > >“I still think that the GUIDE method is difficult to apply to reinforcement learning agent that require long term training (e.g., a few days, atari/procegen/minecraft),”
> > >
> > > >“I have some concerns about GUIDE’s usability for real-world tasks”
> > >
> > > We find these points raised by the reviewer to be puzzling and self-conflicting.
> > > As you have mentioned your concern for real-world applicability, but the environments mentioned by you (e.g., atari/procegen/minecraft) are not real-world tasks. In fact, our bowling task is one of the challenging ones in atari. Both of our find treasures and hide-and-seek are similar to minecraft sub-tasks. Given our setup with partial visual observation, long-horizon planning, and difficult-to-design reward functions without extra info, we believe our tasks are not toy examples. **As stated above, this is evidenced by our results in Fig. 5 where SOTA RL struggles to converge while GUIDE improves them by a large margin.** We believe that this shows the exact potential of GUIDE. It is infeasible to train regular RL on an unrealistic amount of experience, this is exactly the purpose of using real-time human guidance for accelerating agent learning.
> > >
> > > As shown in Fig 5 of our paper, **untrained human participants using GUIDE are able to reach the same level of performance as the RL baseline within half the training time.** We believe these are encouraging results and it shows the potential of real-time human guidance.
> > >
> > > >“sustained manual loads for a few days would still be excessively costly and inconsistent”
> > >
> > > We agree that human feedback is expensive and effectiveness varies across individuals. However, we addressed both of these issues by introducing a simulated feedback provider and conducting the largest scale analysis existing on the effect of individual differences on AI guidance. This has not been done before.
> > >
> > > >“and a few of the toy experiments in the paper don’t support the author’s point of view.”
> > >
> > > We would like to emphasize that RL baselines still struggle on the tasks that the reviewer categorized as “toy examples.” Again, shown in Fig 5, GUIDE surpassed the baseline by a large margin. **Overall, we don’t consider devaluing our contribution simply based on a gap already pointed out by our experiments and us showing strong results in scalability to be appropriate for scientific advancements.** We do not feel the reviewer’s points directly relate to our work.

---

> > > > ### Comment · Reviewer_JtyH · 2024-08-12
> > > >
> > > > I believe the author misunderstood my point. My core opinion is that real-world tasks are usually very complex and cannot be learned in just 10 minutes. Therefore, longer training times and more complex environments pose significant challenges to the GUIDE method mentioned in the paper. Many RLHF methods use environments like Atari or Minecraft for training to demonstrate their effectiveness in complex settings. However, the author has not proven this point, and the simulated feedback reward section lacks sufficient and comprehensive experimental evidence. I am not fully convinced that GUIDE can promote the development of the RLHF community. The feedback method of GUIDE is somewhat novel but useless, it requires a high level of annotator focus and a considerable tolerance for errors, making it infeasible for long-term annotation tasks.

---

> > > > > ### Author Response · Authors · 2024-08-12
> > > > >
> > > > > Thank you for your response. We would like to address all of your concerns and questions below.
> > > > >
> > > > > ----
> > > > > >“I believe the author misunderstood my point. My core opinion is that real-world tasks are usually very complex and cannot be learned in just 10 minutes. Therefore, longer training times and more complex environments pose significant challenges to the GUIDE method mentioned in the paper. Many RLHF methods use environments like Atari or Minecraft for training to demonstrate their effectiveness in complex settings. However, the author has not proven this point, and the simulated feedback reward section lacks sufficient and comprehensive experimental evidence. I am not fully convinced that GUIDE can promote the development of the RLHF community. “
> > > > >
> > > > > We would like to ask for specific clarifications from the reviewer. Though we have emphasized that one of our tasks is a challenging Atari game, and the other two tasks are similar to sub-tasks in Minecraft, the reviewer kept referring to “many RLHF methods use environments like Atari or Minecraft.” To our knowledge, and as stated in our paper, state-of-the-art algorithms with the same settings as ours (real-time human-guided RL) use much simpler environments than ours. In fact, ours is the most challenging benchmark experiment. For instance, Deep Tamer uses bowling, which is included in our experiment, and Deep COACH[1] uses Minecraft (a drastically simplified version that is simply goal navigation in a 10 x 10 grid world with discrete actions). Ours involve continuous control from partial visual observations and competitive settings.
> > > > >
> > > > > If the reviewer is referring to other work that uses Atari or Minecraft but does not fall into our same problem setting, such as preference-based offline setting or imitation learning, we would like to argue that this is not a fair request to ask us to compare or use the same tasks. Our setting is much more challenging – 1) we do not assume any expert level of task execution as most imitation learning algorithms do; 2) our setup aims to tackle novel problems in real-time instead of assuming plenty of time and many offline rollouts or replay buffers for humans to provide feedback. Therefore, we kindly ask the reviewer to point at specific literature that uses the same setting but uses Atari or Minecraft as their comparisons.
> > > > >
> > > > > >“The feedback method of GUIDE is somewhat novel but useless, it requires a high level of annotator focus and a considerable tolerance for errors, making it infeasible for long-term annotation tasks.”
> > > > >
> > > > > We disagree with the reviewer’s statement. We would like to point out that the reviewer’s comments are based on conjecture without evidence.  As stated above, from our results, there is no evidence showing that a continuous feedback interface poses a higher cognitive load than discrete feedback since both require the users to watch the agent continuously, while GUIDE does not require constant switching and clicking on different buttons, nor evidence showing that continuous feedback requires a considerable tolerance for errors. From our 50 human subject studies as the experiment with the largest scale, our results show the opposite. Our continuous feedback mechanism still performs stronger than the discrete feedback baselines by a large margin on all 50 subjects, while the discrete feedback is less robust against individual differences. We have demonstrated the effectiveness of our feedback mechanism in our experiment results, as pointed out above, contrary to the reviewer’s comments that our feedback method is “useless”.
> > > > >
> > > > > ----
> > > > > *references*
> > > > >
> > > > > [1] Arumugam, Dilip, et al. "Deep reinforcement learning from policy-dependent human feedback." arXiv preprint arXiv:1902.04257 (2019).

---

### Official Review · Reviewer_Xw63 · 2024-07-07

**Soundness:** 3
**Presentation:** 2
**Contribution:** 3
**Rating:** 5
**Confidence:** 3

**Summary:**

The paper proposes a new approach to reinforcement learning with human feedback in simple video games. The method relies on continuous human feedback that is provided by the human observer hovering their mouse over a window with a spectrum of positive and negative rewards. Unlike prior approaches, this method converts human feedback directly into a reinforcement learning reward with an added time delay. Moreover, the method includes a model that regresses states into observed human feedback, which allows for simulated human feedback. The effectiveness of the method is demonstrated in three simple games in a human study with 50 participants.

**Strengths:**

1. The authors propose a simple way to incorporate continuous human feedback as a reinforcement learning reward with a constant time delay. The environment reward and human feedback are simply added to form the final reward function.

2. It is demonstrated that human preference can be directly regressed from states and actions to provide simulated human feedback.

3. The authors perform an extensive human study showing the effectiveness of their method. They also correlate the subject’s performance in a cognitive test with their ability to guide an RL agent.

**Weaknesses:**

There are three major unstated assumptions:

1. The delay between an event appearing on the screen and the change in human feedback is constant (Question 1). The authors tune this constant for each environment. But, more complex environments might induce different delays as the human observer might need to think about what they saw.

2. People are able to provide constant feedback (Question 2). This might not be true for more complex environments where certain states might have ambiguous values.

3. The human feedback is Markov (Question 3). This might not be true in more complex games.

## Detailed comments:

* Equation 1 should have Q instead of A. Unless you want to define an advantage function A.
* Equation 2 should have an upper-case Q.
* The term $R_{t+k+1}$ in Equation 2 is not very clear.
* The meaning of “We follow recent advancements in neural architectures and hyperparameter designs” on line 125 is not clear.
* The rest of the paragraph on lines 125 - 127 is superfluous.

## Minor comments:

* Inconsistent spacing between text and citations.
* Calling this approach a “computational framework” might be a bit redundant given the context of the conference.

**Questions:**

1. Is it reasonable to assume constant reaction time of the human observer?

2. How would your method deal with games with more ambiguous state values, such as Montezuma's Revenge?

3. The human feedback regressor $\bar{H}(s, a)$ uses the Markov assumption, but the human feedback might be dependent on their memory of prior states. What if the human feedback is non-Markov?

**Limitations:**

Limitations are partially addressed.

---

> ### Author Rebuttal · Authors · 2024-08-03
>
> We thank the reviewer for their thoughtful comments. We would like to address all of your concerns and questions below with point responses:
>
> ----
> >“There are three major unstated assumptions: The delay between an event appearing on the screen and the change in human feedback is constant (Question 1). The authors tune this constant for each environment. But, more complex environments might induce different delays as the human observer might need to think about what they saw.”
> (Question1: “Is it reasonable to assume constant reaction time of the human observer?”)
>
> We agree that human response time can have various delays. In GUIDE, the advantage of using continual feedback is that it can partially alleviate this issue by eliminating the need for time window association for feedback to state-action pairs. Such a time window is required in previous discrete feedback designs. However, modeling more fine-grained response delay can potentially further improve GUIDE. We leave the exploration of this aspect in future work.
>
> >“People are able to provide constant feedback (Question 2). This might not be true for more complex environments where certain states might have ambiguous values.”
>
> Our assumption is that humans have a preference for any given state-action pair, though the strength of the preference may vary. Continuous-valued feedback can model the strength of preference, e.g., provide a feedback value with a smaller magnitude to reflect weaker preferences. Our experiments suggest that our method surpasses past work on discrete feedback. One future research direction could be empowering the model to learn adaptive changes in human feedback over the course of trajectories or use multiple feedback modalities. We believe our framework can be a strong starting point for incorporating these improvements. We leave such exploration as future work.
>
> >(Question 2: “How would your method deal with games with more ambiguous state values, such as Montezuma's Revenge?”)
>
> Our method of incorporating human feedback can also encourage exploration behaviors which is critical for challenging tasks such as Montezuma’s Revenge. We have observed a similar conclusion in Find Treasure, where human feedback can help explore the environment while the target object is not yet available to the agent, as in Fig. 7.  Our method is orthogonal to approaches that aim to reduce the ambiguity of feedback specifications. Exploring methods to reduce such ambiguity can be an interesting future direction.
>
> >“The human feedback is Markov (Question 3). This might not be true in more complex games.”
> (Question3: “The human feedback regressor $\bar{H}(s, a)$ uses the Markov assumption, but the human feedback might be dependent on their memory of prior states. What if the human feedback is non-Markov?”)
>
> We agree that human feedback may not always be Markov for any task. As we mentioned in Section 5.4, for Find Treasure and Hide-and-Seek, we stacked three consecutive frames as input to both the RL backbone and the feedback regressor. We found this to be sufficient to model human feedback. For more complex tasks, it will be interesting to explore whether incorporating more prior history steps or using a memory-based neural network will improve human feedback modeling.
>
> >“Minor typo and wording issues.”
>
> We thank the reviewer for pointing out some typos and wording issues. We will improve these in the revised version: 1) change A to Q in Equation 1; 2) use upper-case Q in Equation 2; 3) Define $R_{t+k+1}$; 4) list specific designs in line 125 and be specific for lines 125-127; 5) fix spacing in citations; 6) rename computational framework.

---

> ### Author Response · Authors · 2024-08-07
>
> Dear Reviewer,
>
> Thank you again for your detailed review of our paper. We aim to try our best to address all your concerns with our point responses above.
>
> We truly value your feedback. As the end of the rebuttal period is approaching, please feel free to let us know if you have any additional questions or comments. We would be happy to answer them. We look forward to hearing your future thoughts!
>
> Best regards,
>
> Authors

---

### Official Review · Reviewer_sJ5r · 2024-07-12

**Soundness:** 3
**Presentation:** 3
**Contribution:** 2
**Rating:** 6
**Confidence:** 4

**Summary:**

The paper proposes a new framework for human-in-the-loop reinforcement learning, where the human and provide real-time and continuous feedback, and an algorithm where the learning agents uses the human feedback to accelerate policy learning. The paper conducted a user study of 50 subjects to demonstrate the effectiveness of the proposed framework in accelerating policy learning and improving success rates over RL and human-in-the-loop RL baselines. Optionally, a human feedback simulator can also be trained to mimic human feedback after a certain amount of time, reducing the amount of human input.

**Strengths:**

- The paper proposes human-in-the-loop RL where the human can provide real-time continuous feedback, which is a novel paradigm compared to mainstream existing work which focus on discrete feedback signals.
- This work conducted a user study of 50 subjects, which is the largest among relevant works. This is a great contribution in assessing the effectiveness of human-in-the-loop RL.
- The evaluation is done on three challenging tasks, and GUIDE outperform all the baselines by a large margin on the "find treasure" and "high and seek" tasks.
- The paper provides a detailed individual difference characterization by conducting a series of human cognitive tests. Analysis of the human cognitive test data provides meaningful insights. These data can also be very useful in future work.

**Weaknesses:**

- The baselines are generally quite weak. Based on the experiment results, it is unclear whether real-time continuous feedback is necessarily the best way for humans to guide the policy learning. There might be intermediate points on the spectrum of conventional discrete feedback and full continuous feedback that provides the best tradeoff between amount of human input and effectiveness of guiding policy learning.
- Whether the simulated human feedback is helpful is unclear. In both the "bowling" the "find treasure" task, the score does not increase much after switching to simulated feedback. It might be the case that the simulated human feedback only works for tasks where it is straightforward to model the reward.

**Questions:**

- How does human-in-the-loop compare to BC or reward engineering for the same amount of human input?
- How are the three tasks for evaluation chosen, and what are the motivations for such choices?

One motivation for doing human-in-the-loop RL is for tasks where designing a reward function or providing demonstrations are difficult, such as the backflip task. However, for the tasks examined in this paper, there seems to be other simple ways for the RL agent to learn from human input.

**Limitations:**

The paper focused on the final success rate of policy learning, but did not provide sufficient data from the user's perspective. For example, the user study failed to include a survey regarding whether the real-time feedback system feels easier to use than the discrete feedback system.

---

> ### Author Rebuttal · Authors · 2024-08-03
>
> We thank the reviewer for their thoughtful comments. We are glad that the reviewer found our method to be novel, the scale of our human study to be “a great contribution,” and our human analysis to be insightful. We would like to address all of your concerns and questions below with point responses:
>
> ----
> >“The baselines are generally quite weak. Based on the experiment results, it is unclear whether real-time continuous feedback is necessarily the best way for humans to guide the policy learning. There might be intermediate points on the spectrum of conventional discrete feedback and full continuous feedback that provides the best tradeoff between amount of human input and effectiveness of guiding policy learning.”
>
> We agree that it will be interesting to explore the “optimal” feedback mode. Based on feedback from our human participants, the continuous feedback mode was not cognitively demanding, hence a full continuous feedback that enables maximum human input is a straightforward and effective choice. We will leave the exploration of blending discrete and continuous feedback as future work.
>
> >“Whether the simulated human feedback is helpful is unclear. In both the "bowling" the "find treasure" task, the score does not increase much after switching to simulated feedback. It might be the case that the simulated human feedback only works for tasks where it is straightforward to model the reward.”
>
> We agree that the effectiveness of simulated human feedback will vary depending on the complexity of the environment and human feedback. The main purpose of the simulated feedback is to allow continual training without the human trainer and minimize the shift in reward distribution, which is a novel algorithm design. We will leave this interesting investigation of the relationship between task complexity and the effectiveness of simulated feedback as future work.
>
> >“How does human-in-the-loop compare to BC or reward engineering for the same amount of human input?”
>
> It is unclear how to quantify the amount of human input for reward engineering. One advantage of GUIDE is that we do not require human subjects to have prior knowledge of RL. However, reward engineering typically requires domain expert knowledge and experience in designing rewards for tuning RL policies. On the other hand, BC typically assumes expert demonstrations, which also demand more cognitive load from the subjects and expert-level demonstrations. However, for complex tasks, humans may not easily provide high-quality demonstrations while they can still provide an assessment of the agent’s decisions. Therefore, we believe that BC and reward engineering are not suitable comparisons in our case.
>
> >“How are the three tasks for evaluation chosen, and what are the motivations for such choices?”
>
> The bowling task is a classic environment commonly used in prior literature [1, 2, 3]. We included it to compare it with Deep Tamer. Find Treasure is a navigation task motivated by potential real-world applications of human-guided RL, such as search and rescue and disaster response. It also serves as an intermediate task before hide-and-seek, where the target does not move. Hide-and-seek is selected as a representative task involving adversarial competition. Most existing tasks in literature are discrete action tasks with low-dimensional states. We believe that our task selection is a large step forward in investigating the scalability of human-guided RL.
>
> >“One motivation for doing human-in-the-loop RL is for tasks where designing a reward function or providing demonstrations are difficult, such as the backflip task. However, for the tasks examined in this paper, there seems to be other simple ways for the RL agent to learn from human input.”
>
> Despite reward engineering that could exist for these tasks, our setting is different. Designing rewards requires expert knowledge and RL training experience. However, our setting does not assume such expert experience in our human subjects. Moreover, designing dense rewards for these tasks requires extra information that is not available to our agents, such as the position of the agent, the target object location (our environment only provides partial visual inputs where target objects may not be visible), and the location of the adversarial agent (our environment only provides partial visual inputs where the other adversarial agent are not always visible). Therefore, our setting does fulfill the suggestions from the reviewer where the reward function is difficult to design without such extra experience or extra exposed information.
>
> >“The paper focused on the final success rate of policy learning, but did not provide sufficient data from the user's perspective. For example, the user study failed to include a survey regarding whether the real-time feedback system feels easier to use than the discrete feedback system.”
>
> We would like to clarify that our work also quantifies individual differences and their impact on guided AI performance, as in our cognitive studies in Fig. 5 and 7. Surveys about user experience can be subjective. One possible solution is to quantify physiological signals to measure their cognitive load and stress level, combined with surveys. We leave such exploration in future work. Though we did not have a formal survey for this question, verbal feedback from our subjects suggests that most of them did not find the continuous feedback interface cognitively demanding.
>
> ----
> *references*
>
> [1] Warnell, Garrett, et al. "Deep tamer: Interactive agent shaping in high-dimensional state spaces." Proceedings of the AAAI conference on artificial intelligence. Vol. 32. No. 1. 2018.
>
> [2] Park, Sung-Yun, Seung-Jin Hong, and Sang-Kwang Lee. "Human Interactive Learning with Intrinsic Reward." 2023 IEEE Conference on Games (CoG). IEEE, 2023.
>
> [3] Xiao, Baicen, et al. "Fresh: Interactive reward shaping in high-dimensional state spaces using human feedback." arXiv preprint arXiv:2001.06781 (2020).

---

> > ### Comment · Reviewer_sJ5r · 2024-08-08
> >
> > I thank the authors for addressing my questions. I increased the confidence score to 4.
> >
> > Considering the overall contributions and limitations of the paper, I will keep my other ratings.

---

> > > ### Author Response · Authors · 2024-08-08
> > >
> > > We appreciate your positive feedback! Thank you!

---

> ### Author Response · Authors · 2024-08-07
>
> Dear Reviewer,
>
> Thank you again for your detailed review of our paper. We aim to try our best to address all your concerns with our point responses above.
>
> We truly value your feedback. As the end of the rebuttal period is approaching, please feel free to let us know if you have any additional questions or comments. We would be happy to answer them. We look forward to hearing your future thoughts!
>
> Best regards,
>
> Authors

---

### Official Review · Reviewer_fsCr · 2024-07-13

**Soundness:** 2
**Presentation:** 3
**Contribution:** 2
**Rating:** 4
**Confidence:** 2

**Summary:**

This paper proposes a new framework, GUIDE, for learning from continuous human feedback in complex decision making domains with continuous action spaces. By framing human feedback as a state-action value function, the framework proposes to learn this function and to combine it additively with the (generally sparse) reward coming from the environment. The feedback is collected in a continuous fashion by asking participants to move their mouse up or down to indicate higher or lower feedback values. In a user study, the paper finds that training agents with this type of feedback yields better performing agents than baselines. After assessing participants in a suite of cognitive tests, it finds that participants that score higher on the cognitive tests trained better agents.

**Strengths:**

The paper introduces an interesting new way of collecting continuous human feedback, and shows significant improvement in two out of three tasks considered. The use of cognitive tests as part of the user study is interesting, and uncovers insightful correlation between subject performance and their cognitive test scores.

**Weaknesses:**

The assumptions regarding what human feedback represents do not seem consistent between section 3 and 4 (see Questions). Further, the treatment of the feedback collection is rather simple (added to environment reward function) and, especially if it does represent a signal regarding the future value of state-action pairs, heuristic. Relative to Tamer and Deep Tamer, which treated human feedback more consistently by using it directly as a proper state-action value function, this paper feels like a regression on that front.

The implementation of the c-DeepTamer baseline raise a number of questions (see Questions), which shake my confidence in it as a baseline, or as a proper representative for how well Deep Tamer should perform here.

**Questions:**

In 3.2 you propose to extend Deep Tamer by using the human feedback estimator F(s,a) as a critic within an actor-critic framework. Thus, you treat human feedback here as an estimate of the future value of each state-action pair (a Q-value). In 4.3, you propose to use the human feedback signal as a reward, to be added to the environment reward. Does this mean you make a different assumption regarding human feedback in 4.3. than in 3.2? Or do you propose to use feedback on future value as a reward?

I am confused by your claim that Deep Tamer uses DQN. Looking at algorithm 1 in the Deep Tamer paper, it’s policy greedily chooses the action that maximizes the learnt human value function H(s,a). I do not see any use of DQN.

Did your c-DeepTamer baseline receive the same (sparse) environment reward as GUIDE? And if so, how did you use this reward signal in conjunction with a critic that is trained from human feedback only?

Why was the training of c-DeepTamer stopped after 5/10 minutes? Would it not be possible to continue training it using simulated human feedback (similar to GUIDE) or without human feedback?

Within the user study, did participants get time to familiarize themselves with each environment before data recording began? I imagine participants would have needed some time to figure out how to solve each task.

Is equation (3) correct? What is the purpose of he additional $- F(s, A(s))$ term?

**Limitations:**

Limitations are discussed in the conclusion.

---

> ### Author Rebuttal · Authors · 2024-08-03
>
> We thank the reviewer for their thoughtful comments. We would like to address all of your concerns and questions below with point responses:
>
> ----
> >“The assumptions regarding what human feedback represents do not seem consistent between section 3 and 4 (see Questions). ”
>
> We would like to clarify that c-DeepTamer and GUIDE use human feedback differently. c-DeepTamer, described in sec 3.2, is a baseline algorithm. It is an upgraded version of the original Deep Tamer[1], and follows the original Deep Tamer formulation to use human feedback as the myopic state-action value function. We did not modify how Deep Tamer uses human reward as a fair comparison. In GUIDE, we instead use human feedback as an additional immediate reward to the sparse environment reward.
>
> >“Further, the treatment of the feedback collection is rather simple (added to environment reward function) and, especially if it does represent a signal regarding the future value of state-action pairs, heuristic. Relative to Tamer and Deep Tamer, which treated human feedback more consistently by using it directly as a proper state-action value function, this paper feels like a regression on that front.”
>
> Tamer and Deep Tamer can treat human feedback directly as a state-action value function because they ignore the sparse environment feedback. However, we believe that it is still useful to leverage the sparse environment feedback for RL training. In GUIDE, Our design choice to use human feedback as an additional reward function is based on this intuition to integrate both types of feedback together.
>
> >“In 3.2 you propose to extend Deep Tamer by using the human feedback estimator F(s,a) as a critic within an actor-critic framework. Thus, you treat human feedback here as an estimate of the future value of each state-action pair (a Q-value). In 4.3, you propose to use the human feedback signal as a reward, to be added to the environment reward. Does this mean you make a different assumption regarding human feedback in 4.3. than in 3.2? Or do you propose to use feedback on future value as a reward?”
>
> Yes, we use human feedback differently in c-Deep Tamer and GUIDE. In Deep Tamer, human feedback is treated as a myopic state-action value. In GUIDE, we treat human feedback as a reward function, which we believe is a more appropriate and effective approximation, as evidenced by our experimental results. Moreover, this design choice makes it more natural to incorporate sparse environment rewards with the human reward in practice.
>
> >“I am confused by your claim that Deep Tamer uses DQN. Looking at algorithm 1 in the Deep Tamer paper, it’s policy greedily chooses the action that maximizes the learnt human value function H(s,a). I do not see any use of DQN.”
>
> Deep Tamer can be seen as DQN with $\gamma=0$, where the target state-value function at the current time step is directly assigned by humans. We realize that this phrase may cause confusion. We will remove the DQN claim in the revised paper and instead describe as Deep Tamer learns a reward model to regress human feedback, which is then used for greedy action selection.
>
> >“Did your c-DeepTamer baseline receive the same (sparse) environment reward as GUIDE? And if so, how did you use this reward signal in conjunction with a critic that is trained from human feedback only?”
>
> c-DeepTamer only upgrades Deep Tamer to handle continuous actions, with extensive recent neural network designs and hyperparameter tuning. Following Deep Tamer, it does not use an environment reward. As stated above, Deep Tamer treats human feedback as a myopic state-action value, which is not really a reward, making it impractical to add an environment reward to it. Changing the reward setup in Deep Tamer will make it unfair to compare. One contribution of GUIDE is to provide a formulation that uses both human reward and sparse environment reward.
>
> >“Why was the training of c-DeepTamer stopped after 5/10 minutes? Would it not be possible to continue training it using simulated human feedback (similar to GUIDE) or without human feedback?”
>
> We strictly follow the original Deep Tamer implementation which does not handle continual training. In the original Deep Tamer, the model weights are only updated every time when a new human feedback signal is received (see Algorithm 1, line 5 in the Deep Tamer paper[1]). When there is no human feedback available, Deep Tamer will not be improved.
>
> Both c-Deep Tamer and GUIDE are trained with the same amount of human guidance time. In fact, one contribution of GUIDE is to provide a mechanism to simultaneously train a simulated feedback model for continual training once human guidance is not available. We would like to clarify that this is our contribution in GUIDE instead of an inheritance from Deep Tamer.
>
> >“Within the user study, did participants get time to familiarize themselves with each environment before data recording began? I imagine participants would have needed some time to figure out how to solve each task.”
>
> Yes. Before each experiment, we showed the participants a short video introducing the goal of each game and a quick demonstration of how each game is played and how the feedback interface can be used.
>
> >“Is equation (3) correct? What is the purpose of the additional -F(s, A(s)) term?”
>
> Mathematically, the equation (3) is correct. The additional term -F(s, A(s)) is the actor loss, which tries to maximize the feedback value predicted by the critic network, given an action selected by the actor A(s). We recognize that the presentation may cause confusion since the actor and critic loss only affect their corresponding networks separately. We will clarify this by separating these two terms and indicate model updates in our revised paper.
>
> ----
> *references*
>
> [1] Warnell, Garrett, et al. "Deep tamer: Interactive agent shaping in high-dimensional state spaces." Proceedings of the AAAI conference on artificial intelligence. Vol. 32. No. 1. 2018.

---

> ### Author Response · Authors · 2024-08-07
>
> Dear Reviewer,
>
> Thank you again for your detailed review of our paper. We aim to try our best to address all your concerns with our point responses above.
>
> We truly value your feedback. As the end of the rebuttal period is approaching, please feel free to let us know if you have any additional questions or comments. We would be happy to answer them. We look forward to hearing your future thoughts!
>
> Best regards,
>
> Authors

---

> ### Comment · Reviewer_fsCr · 2024-08-12
>
> Thank you for your detailed response. Unfortunately I continue to see two weaknesses in the paper.
>
> I like the proposed mechanism for giving feedback. It is elegant and to my knowledge novel. Reviewer JtyH has a point by saying that complex RL tasks will need more significant amounts of feedback, but in my view any interactive reward learning method would. This could have been explored more in the paper. The idea of combining environment reward with human feedback, however, cannot be counted as a novel contribution considering earlier work in this area [A,B]. Overall, the technical contribution is thus limited.
>
> A very solid set of human subject experiments would have been able to compensate for this, but in my view the experiments presented at this point are not quite complete enough. The c-DeepTamer baseline is artificially limited. My best understanding is that each bit of feedback was used only once in a gradient update, contrary to standard ML practice (please correct me if I'm wrong). Further, c-DeepTamer is not able to make use of the environment's reward, raising questions about whether it is a fair baseline in the first place. Why not use a baseline like the [A,B] that could use the environment's reward? The paper could also have been made much stronger by comparing to alternative interactive methods for learning from humans, including learning from preferences [C] and learning from demonstrations (e.g. [B]).
>
> I appreciate that the authors have clarified some of my concerns, and will raise my score accordingly, but I still feel that the paper falls short of the acceptance threshold.
>
> [A] Xiao, Baicen, et al. "FRESH: Interactive Reward Shaping in High-Dimensional State Spaces using Human Feedback." Proceedings of the 19th International Conference on Autonomous Agents and MultiAgent Systems. 2020.
> [B] Brys, Tim, et al. "Reinforcement learning from demonstration through shaping." Proceedings of the 24th International Conference on Artificial Intelligence. 2015.
> [C] Christiano, Paul F., et al. "Deep reinforcement learning from human preferences." Advances in neural information processing systems 30 (2017).

---

> > ### Author Response · Authors · 2024-08-12
> >
> > Thank you for your response. We are glad that the reviewer found the feedback mechanism to be elegant and novel. We would like to address all of your concerns and questions below.
> >
> > ----
> > > “I like the proposed mechanism for giving feedback. It is elegant and to my knowledge novel. Reviewer JtyH has a point by saying that complex RL tasks will need more significant amounts of feedback, but in my view any interactive reward learning method would. The idea of combining environment reward with human feedback, however, cannot be counted as a novel contribution considering earlier work in this area [A,B]. Overall, the technical contribution is thus limited.”
> >
> >
> > We agree that prior work has also used human feedback for reward shaping. However, we believe that our main contribution is using such a method to ground real-time dense continuous feedback and conduct large-scale human experiments to verify it. We will add relevant discussion of this literature in our revised manuscript.
> >
> >
> > > “in my view the experiments presented at this point are not quite complete enough. The c-DeepTamer baseline is artificially limited. My best understanding is that each bit of feedback was used only once in a gradient update, contrary to standard ML practice (please correct me if I'm wrong).”
> >
> >
> > To maintain a fair comparison, c-DeepTamer strictly follows Deep TAMER in the way gradients are updated. As described in the original Deep TAMER paper Algorithm 1, the gradient updates happen:
> >
> > (a) whenever the human provides new feedback, using the new feedback information as the data sample; and
> >
> > (b) at a fixed rate, using data sampled from the replay buffer.
> >
> > We believe that a stronger solution would be able to update the network as standard ML practice, which is exactly one of our contributions beyond the original DeepTamer work. However, if we modify the original DeepTamer work to include our contribution, our experiments will become an ablation study of our own method instead of comparing it with prior work. Considering the expensive 50 human studies, which already cost 2 hours per subject and $1,000, running ablations is unfortunately not feasible. Our choice is to focus on comparing the previous literature in the same setting as our problem.
> >
> >
> > > “Further, c-DeepTamer is not able to make use of the environment's reward, raising questions about whether it is a fair baseline in the first place. Why not use a baseline like the [A,B] that could use the environment's reward? The paper could also have been made much stronger by comparing to alternative interactive methods for learning from humans, including learning from preferences [C] and learning from demonstrations (e.g. [B]).”
> >
> >
> > We agree that additional baselines would be interesting to compare with. However, experiments involving real-time human interactions are costly to run. Our current 50 human experiments include three environments, two algorithms, cognitive tests, and initial and ending setup, as traditional human experiments already cost nearly 2 hours. The total cost of the experiments is $1,000. Longer experiment sessions will not only be more costly but will also affect human feedback quality. Under the time constraint, we chose the baseline that had the closest setting to ours. Deep TAMER is the state-of-the-art in real-time human-guided RL through scalar feedback of state-action pairs. [A] is not real-time, as the human operator provides feedback to trajectories sampled from a replay buffer. They also only provide feedback to actions or states instead of state-action pairs. As the reviewer has mentioned, [B] learns from demonstration, and [C] learns from preference, which are different settings from learning from feedback.
> >
> > We would like to clarify that our work did not argue that real-time continuous human feedback is the optimal modality to provide feedback. In particular, we did not argue that this modality is better or can replace demonstration or preference learning. For demonstration, it has a different assumption that humans may be experts in the tasks, and this does not fall into our problem setting. In fact, we observe that humans typically have difficulty performing very well in our challenging hide-and-seek task where things change very fast, and the observations are partial. Preference learning is offline, which assumes multiple rollouts are provided, and humans do not need to react in real-time. It is often applied to the whole trajectories instead of state-action pairs. Our real-time continuous human feedback proposes a different problem setup and can be an alternative modality. However, we leave the exploration of multiple modalities as future work. Therefore, from both the perspective of financial, time and human costs and the problem-setting perspective, we did not compare the baselines from different modalities or problem settings. Our experiments indeed show significant improvement over real-time discrete feedback, which supports our claims.

---

### Decision · Program_Chairs · 2024-09-25

**Decision:**

Accept (poster)

**Comment:**

This paper presented a method that the reviewers found novel / interesting, and that overall the experiments supported the papers claims. However, the paper elicited a wide variety of concerns and some spirited debate.

# Concerns of the reviewers who rated the paper the lowest
## Reviewer fsCr (rating=4)
This reviewer had some concerns about the overall technical novelty, and said
> "a very solid set of human subject experiments would have been able to compensate for this ... but ... the experiments presented at this point are not quite complete enough".

They pointed to concerns about the validity of the c-DeepTamer baseline both in terms of whether it is appropriate and whether the number of gradient steps used is valid, which the authors rebutted by stating that their baseline
>  "strictly follows Deep TAMER in the same way gradients were updated" and "We believe that a stronger solution would be able to update the network as standard ML practice, which is exactly one of our contributions beyond the original DeepTamer work. However, if we modify the original DeepTamer work to include our contribution, our experiments will become an ablation study of our own method instead of comparing it with prior work.".

I think the authors rebutted this point sufficiently. Furthermore, this reviewer suggested [A,B] as additional baselines, which the authors responded to partly by saying "experiments involving real-time human interactions are costly to run." I understand this point, and agree that it weakens the paper (regardless of the cost to run the experiments). The paper did not claim to be superior to prior work in its "key novelties" paragraph or experimental section, so this weakness doesn't undermine the main contributions as stated in those sections, yet the abstract _does_ state
>  "up to 32% increase in success rate compared to state-of-the-art baselines"

which is ambiguous and potentially misleading, as it suggests _all relevant baselines_ (including [A,B]) were run. Therefore, while I think this criticism is valid, it doesn't undermine the key novelties as stated in the majority of the paper, yet it does undermine a novelty stated in the abstract. Thus, I recommend the paper be accepted and the abstract be revised to more accurately and less ambiguously state the performance improvement, to be consistent with the remaining claims of the paper.

## Reviewer JtyH (rating=3)
This reviewer and the authors got into a spirited debate. Near the end of this discussion, the reviewer said
> “The feedback method of GUIDE is somewhat novel but useless, it requires a high level of annotator focus and a considerable tolerance for errors, making it infeasible for long-term annotation tasks.”

The authors responded to this by stating
> "the reviewer’s comments are based on conjecture without evidence"
> "Our continuous feedback mechanism still performs stronger than the discrete feedback baselines by a large margin on all 50 subjects, while the discrete feedback is less robust against individual differences."

Thus, the authors rebutted this point satisfactorily, in my judgement. This reviewer also stated
> "My core opinion is that real-world tasks are usually very complex and cannot be learned in just 10 minutes".

The authors responded partly by stating that the task attempted was challenging. I think both of these claims -- the authors's and reviewer's -- are hard to agree with because they are not backed by quantitative evidence. While I agree with the reviewer that evaluation on longer-term tasks would be an interesting direction for future work (and a stronger proof of efficacy), I believe it is not necessary for this paper to be acceptable, as it doesn't conflict with the paper's main claims. The authors also stated
> "1) we do not assume any expert level of task execution as most imitation learning algorithms do; 2) our setup aims to tackle novel problems in real-time instead of assuming plenty of time and many offline rollouts or replay buffers for humans to provide feedback. "

I think the paper needs to clarify these assumptions and differences, e.g L185-186 states their approach allows for "capitalizing on human flexibility and expertise", which contradicts (1) above.

Given the somewhat strong consensus on the novelty and that the paper's claims seem reasonably supported by the evidence presented, I recommend acceptance.